# A genome-wide association study in Indian wild rice accessions for resistance to the root-knot nematode *Meloidogyne graminicola*

**Alkesh Hada[1], Tushar K. Dutta[1]\*, Nisha Singh[2], Balwant Singh[2], Vandna Rai[2], Nagendra K. Singh[2], Uma Rao[1]\***

**1** Division of Nematology, ICAR-Indian Agricultural Research Institute, New Delhi, India, **2** ICAR-National Institute for Plant Biotechnology, New Delhi, India

\* umarao@iari.res.in (UR); nemaiari@gmail.com (TKD)

**Data Availability Statement:** All relevant data are within the paper and its Supporting Information files.

**Funding:** UR received a grant. Grant number: BT/PR18924/COE/34/48/2017 Funder full name:

## Abstract

Rice root-knot nematode (RRKN), *Meloidogyne graminicola* is one of the major biotic constraints in rice-growing countries of Southeast Asia. Host plant resistance is an environmentally-friendly and cost-effective mean to mitigate RRKN damage to rice. Considering the limited availability of genetic resources in the Asian rice (*Oryza sativa*) cultivars, exploration of novel sources and genetic basis of RRKN resistance is necessary. We screened 272 diverse wild rice accessions (*O. nivara*, *O. rufipogon*, *O. sativa* f. *spontanea*) to identify genotypes resistant to RRKN. We dissected the genetic basis of RRKN resistance using a genome-wide association study with SNPs (single nucleotide polymorphism) genotyped by 50K "OsSNPnks" genic Affymetrix chip. Population structure analysis revealed that these accessions were stratified into three major sub-populations. Overall, 40 resistant accessions (nematode gall number and multiplication factor/MF < 2) were identified, with 17 novel SNPs being significantly associated with phenotypic traits such as number of galls, egg masses, eggs/egg mass and MF per plant. SNPs were localized to the quantitative trait loci (QTL) on chromosome 1, 2, 3, 4, 6, 10 and 11 harboring the candidate genes including NBS-LRR, Cf2/Cf5 resistance protein, MYB, bZIP, ARF, SCARECROW and WRKY transcription factors. Expression of these identified genes was significantly ($P < 0.01$) upregulated in RRKN-infected plants compared to mock-inoculated plants at 7 days after inoculation. The identified SNPs enrich the repository of candidate genes for future marker-assisted breeding program to alleviate the damage of RRKN in rice.

## Introduction

Cereals occupy the major share in global food supply in terms of production, acreage and source of nutrition. Rice as a staple food crop meets the nutrient demand of at least two-thirds of the global population ([1]; http://www.ricepedia.org/). However, the yield of rice is substantially afflicted by several biotrophic pathogens including rice root-knot nematode (RRKN), *Meloidogyne graminicola*. This sedentary endoparasite occurs in a wide range of rice-based

Department of Biotechnology, Govt. of India URL of funder website: http://dbtindia.gov.in/ IARI Outreach Project (IARI:ORP:NEM:09:04) The funders had no role in study design, data collection and analysis, decision to publish, or preparation of the manuscript.

**Competing interests:** The authors have declared that no competing interests exist.

production systems in South and Southeast Asia, including lowland, upland, rainfed, irrigated, deep-water as well as aerobic rice, and causes economic yield loss of rice [2–7]. In the vascular bundle, near the root tip, RRKN induces a feeding site (3–5 giant cells) that serves as the permanent nutrient sink for the nematode. Typical hook-shaped galls are formed around the giant cells that affect water and nutrient translocation of plant and in turn plant growth and several yield-contributing traits (such as root and shoot weight/length, tiller number, panicle/ spikelet length, grain weight and percentage filled grains) are impaired [3–6, 8, 9].

In order to mitigate the loss by RRKN in rice, use of chemical nematicides, crop rotation (with mung bean, mustard and sesame as poor hosts) and continuous flooding are the recommended practices [3, 6, 7]. Although prolonged flooding may reduce the RRKN populations by inhibiting the second-stage juveniles (J2) to invade rice roots, increasing scarcity of water for agricultural use may render this practice impractical to adopt. Rotation with other crops during the growing season may incur additional costs to bear for the small-scale rice farmers who constitute a major part of the farming community in Asia. Considering the continual withdrawal of nematicides from the market due to its residual effect on the environment, growing resistant/tolerant rice cultivars offer an economically and environmentally viable option to manage RRKN problems in rice.

Resistance to RRKN has been unraveled in *Oryza longistaminata*, *O. glaberrima* (African rice) and even in *O. sativa* (Asian rice). Nevertheless, very few of these are truly resistant and majority of Asian rice genotypes are susceptible to *M. graminicola* [9, 10]. Efforts were made to introgress RRKN resistance from *O. glaberrima* to *O. sativa*, however, interspecific progenies did not exhibit the same degree of resistance documented with *O. glaberrima* [11, 12]. Resistance to RRKN in rice is reportedly quantitative in nature and governed by several genes with additive effect. Five QTLs (quantitative trait loci, in chromosomes 1, 2, 6, 7 and 9) were found to be associated with root galling by RRKN in RILs (recombinant inbred lines) derived from the cross of *O. sativa* accessions 'Bala' and 'Azucena' [13]. Jena et al. [14] reported QTLs associated with number of galls and eggs per root system on chromosomes 1 and 3 in RILs derived from the cross of *O. sativa* accessions 'Annapurna' and 'Ramakrishna'. Likewise, QTLs for resistance and tolerance to RRKN and other yield-contributing traits (such as root and shoot weight, plant height, % filled grains) were mapped on different chromosomal regions in RILs derived from different *O. sativa* accessions including 'IR64', 'IR78877–208-B-1-2' and *O. glaberrima* accession 'CG14' [7, 9].

Genome-wide association study (GWAS) offers a viable strategy to complement traditional bi-parental linkage mapping in order to discern the genetic basis of trait variation [15]. Compared to classical bi-parental QTL mapping, GWAS takes less time to dissect quantitative traits at higher resolution because of the greater recombination rate and distribution of SNP (single nucleotide polymorphism) markers across the genome of natural plant populations. Additionally, prior knowledge of genotype pedigree/crosses is not required in GWAS as bi-allelic SNP markers can be used to estimate the population structure [16–18]. To date, only in a few studies, GWAS has been used to identify novel QTLs for phytonematode resistance or susceptibility in different plants including Arabidopsis [19, 20] wheat [21], soybean [22] and rice [23]. QTLs associated with root galling were mapped to the chromosomes 1, 3, 4, 5, 11 and 12 in a global panel of Asian rice [23]. Nevertheless, additional studies on geographically different rice panels (such as wild rice accessions that harbor greater genetic diversity) will enrich the resistant gene pool which can be deployed in future rice breeding programs. Due to ever increasing population pressure and rapid urbanization wild rice resources are depleting at an alarming rate. The Indo-Burma region is considered the biodiversity hotspot of wild rice populations [24, 25].

In the present study, we investigated the genetic basis of RRKN resistance in wild rice accessions (constituting *O. nivara* (annual plant), *O. rufipogon* (perennial plant) and the intermediate *O. sativa* f. *spontanea* types) from India using GWAS with a three-pronged approach–(1) screening nematode resistant accessions among 272 diverse rice germplasm, (2) identifying genomic regions significantly associated with RRKN resistance via SNP markers genotyped by 50K "OsSNPnks" genic Affymetrix chip, and (3) expression analysis of candidate genes in nematode-infected plant tissues by reverse-transcription quantitative PCR (RT-qPCR).

## Materials and methods

### Plant materials

A set of 272 diverse genotypes were used in the present study. The set included 270 wild rice accessions (*O. nivara*/*O. rufipogon*/*O. sativa* f. *spontanea*) collected from different agro-climatic zones of India; 156 from Mid Gangetic Plains (MGP), 14 from Upper Gangetic Plains (UGP), 12 from Lower Gangetic Plains (LGP), 20 from West Himalayan Regions (WHR), 9 from East Himalayan Regions (EHR), 22 from West Coastal Plains (WCP), 3 from East Coastal Plains (ECP), 5 from East Plateau Hills (EPH), 23 from Gujarat Plains and Hills (GPH), and 6 from Island Regency (IR). As reference genotypes, *O. sativa* elite cultivars 'Pusa 1121' (*indica*) and 'Taipei 309' (*japonica*) were used. Details of the rice accessions used in the current study are provided in S1 Table and the online database at http://nksingh.nationalprof.in:8080/iwrdb/index.jsp.

### Nematode culture

A pure culture of an Indian isolate of *M. graminicola* Golden & Birchfield was maintained on rice (*O. sativa* cv. Pusa 1121) in a greenhouse. Egg masses were collected from galled roots via sterilized forceps and were kept for hatching on a double-layered tissue paper supported on a mould sieve of wire gauze in a Petri dish containing distilled water. Freshly hatched second stage juveniles (J2s) were used for subsequent experiments.

### Rice accessions screening assay

Seeds of different rice accessions were surface-sterilized via 70% ethanol prior to soaking overnight in distilled water. Seeds were germinated in Petri dish containing wet filter paper in a growth chamber at $28 \pm 2°C$ and 4–5 days old seedlings were used for infection bioassays in Pluronic gel medium (PF-127, Sigma-Aldrich) as described previously [26, 27]. Briefly, 60 ml of 23% PF-127 was poured into $150 \times 20$ mm Petri dish containing 7–9 uniformly distributed seedlings of identical genotype at $<15°C$ and approximately 30 RRKN J2s were inoculated at the root tip of each seedling by a pipette tip. The gel was set at room temperature and Petri plates were incubated at $28°C$ and 14: 10 h light: dark photoperiod in a growth chamber (light level– 300 $\mu$mol m$^{-2}$ s$^{-1}$). At 16 dpi, plantlets were extracted from PF-127 by placing the dishes in an ice bath as the gel liquefies below $15°C$. Roots were stained with acid fuchsin [28], galls were dissected under the microscope, and the number and stage of penetrated nematodes were recorded. Photomicrographs were obtained in a Zeiss Axiocam MRm microscope. Nematode multiplication factor (MF) ratio was calculated [(number of egg masses × number of eggs per egg mass) ÷ nematode inoculum level] to determine the reproductive potential of RRKN in different rice accessions. For each accession three plates were included and the experiment was repeated at least twice.

## Host response of 40 rice accessions to RRKN infection in soil and PF-127

After initial screening of 272 accessions in PF-127 medium, 40 accessions were selected (based on lower counts ($< 2$) of gall and MF value) for further screening based on root galling, total endoparasite counts and MF. Plug trays with 50 wells [$45 \times 45 \times 35$ mm (length × width × height)] were filled with autoclaved soil (collected from IARI rice field) and two seedlings of identical genotype were sown per well. Seedlings were thinned to one healthy seedling in each well at 10 days. Five days later, plantlets were transferred to 18 well [$80 \times 80 \times 70$ mm (length × width × height)] trays containing 200 g soil in each well and each plantlet was inoculated with 200 J2 of *M. graminicola*. The experiment was carried out in a growth chamber at 28˚C, 60% humidity and 14: 10 h light: dark photoperiod (light level– 300 µmol m$^{-2}$ s$^{-1}$). Plants were regularly watered to field capacity and fertilized with Yoshida's nutrient solution [23]. In order to avoid the secondary galling by RRKN, the experiment was terminated at 16 dpi as *M. graminicola* completes its life cycle in 19 days at 24–30˚C in well-drained soil [3, 5, 6]. In PF-127 medium, *M. graminicola* completes its life cycle in 15 days [26, 27]. Plants were carefully harvested, roots were washed in water, galls and other parameters were observed under the microscope. For each accession 12 plants were screened and the experiment was repeated at least twice.

Further, for stringent screening of 40 accessions, a higher nematode inoculum (~2000 J2s) was applied in PF-127 in Petri plates ($150 \times 20$ mm) containing 50–55 seedlings of identical genotypes. Dishes were incubated, plantlets were harvested at 16 dpi and nematode infection potential was assessed as described above.

## Genome-wide association analysis

The set of 272 rice accessions were genotyped using in-house developed "OsSNPnks" 50K genic Affymetrix chip [29, 30] containing a total of 50,051 high-quality SNPs. Alleles for specific SNPs were detected through Axiom Analysis Suite version 2.0 (https://media.affymetrix.com/). SNP density plot was created using R package CM-plot. GWAS was performed with the rice accessions based on five important diverse nematode phenotypic traits such as number of galls, endoparasites, egg masses, eggs per egg mass and MF. A mixed linear model (MLM) was used to infer the association between each SNP and the phenotype, where markers and population structure (Q matrix) effects were fixed and the genotypic effect (K matrix) was random. The MLM was run in TASSEL v.5.2 (http://www.maizegenetics.net), where the additive genetic and residual variance components of the random factors were re-estimated for each SNP. TASSEL 5.2 computed the log likelihoods of the null and alternative models, and the effect of a SNP with its standard error. A *p*-value of $< 0.0001$ (for traits gall, egg mass and MF) with minor allele frequency (MAF) value of $>5\%$ was used as the threshold for significance analysis of marker-trait associations. A *p*-value of $< 0.001$ was used as the threshold for trait eggs per egg mass. To evaluate the genetic diversity across these diverse rice accessions, a phylogenetic tree was constructed by using the haplotype-based neighbour-joining algorithm, and the tree was visualized with FigTree v.1.4.0 [31].

STRUCTURE v.2.3.4 was employed to determine the genetic subpopulations among 272 accessions using a model-based Bayesian clustering method. Ten independent runs were performed setting the hypothetical number of expected populations (K) range from 1–10. The data were processed with burn-in period and replication values of 50,000 each. The number of distinct subpopulations signify the optimal K value which was determined by estimating natural logarithm of the probability of fit (LnP(D)) obtained in the STRUCTURE output, whilst ∆K indicates the rate of change in LnP(D) between successive K values [32]. The Evanno plot was generated in Structure Harvester v.6.0 [33] to determine the peak ∆K values.

Additionally, pairwise linkage disequilibrium (LD) between selective SNP markers was calculated using squared correlation coefficient ($r^2$) between two markers. Graphical LD decay was imputed by GAPIT R package. The broad-sense heritability ($h^2$) of phenotypic traits was estimated as $h^2 = V_G/(V_G+V_E)$, where $V_G$ is genotypic variance and $V_E$ is residual variance. An additive ANOVA model was constructed to determine the phenotypic variation captured by the each set of SNPs.

## Candidate gene prediction

Sequences flanking the SNP markers by 200 kb were selected via rice Pseudomolecule version 7 in the database Rice Genome Annotation Project (RGAP; http://rice.plantbiology.msu.edu/). *In silico* functional annotation of candidate genes was performed (BLASTp, BLAST2GO, gene ontology, Interpro and PFAM hits) to analyze the putative function of genes associated to RRKN or pathogen resistance with reference to previously published literature. Transposon or retrotransposon-related hits were excluded from the analysis.

## RT-qPCR analysis

Expression analysis of the candidate genes was performed on the uninfected and infected root samples of Pusa 1121 at 7 dpi (J2s inoculated in PF-127 medium as described above). Total RNA was extracted from the snap frozen (in liquid nitrogen) whole root systems using NucleoSpin total RNA kit (Macherey-Nagel, Germany) by following the manufacturer's protocol. RNA quantity and quality were assessed in Nanodrop ND-1000 spectrophotometer (Thermo Fisher Scientific) and 2% (w/v) high resolution MetaPhor agarose gel (S1 Fig). First-strand cDNA was synthesized from the total RNA (1 µg) using a SuperScript VILO cDNA synthesis kit (Invitrogen) by following the manufacturer's instructions. Samples were analyzed by RT-qPCR using SYBR Green Supermix kit (Eurogentec) in a Realplex[2] thermal cycler (Eppendorf). To quantify the expression level of candidate genes, specific gene primers were used (S2 Table). cDNA matching actin gene of *O. sativa* was amplified as a reference for constitutive expression using primers listed in S2 Table. Reaction mixture (10 µl) for each sample consisted of 5 µl SYBR Green PCR Master mix (Eurogentec), 750 nM of each primer and 1.5 ng cDNA. RT-qPCR cycling conditions and melt curve program were followed as previously described [26, 27]. At least three biological and three technical replicates were used for each sample. Quantification cycle (Cq) values were imported from Realplex[2] software (Eppendorf). The relative quantities of each candidate gene transcript in the samples were calculated via $2^{-\Delta Cq}$ [ΔCq = (Cq for candidate gene)–(Cq for actin)] method [34]. The stability of the expression of actin in different experimental conditions (leaf, root, stem and seed tissue of resistant and susceptible genotypes in uninfected and nematode infected condition) in terms of reaction efficiency, $R^2$ and Cq values are demonstrated in S2 Fig.

## Statistical analysis

Data of the bioassay experiments were normalized using generalized linear model and subjected to two-way analysis of variance (ANOVA) in SAS software (version 14.1). Relative gall numbers, egg masses, MF etc. in different accessions were compared via post-hoc Tukey's honest significant difference test (HSD) at $P < 0.01$. For qRT-PCR data, significant differential expression of candidate genes between uninfected and infected plants was compared by post-hoc Tukey's HSD test ($P < 0.01$).

## Results

### Host response of 272 rice accessions to RRKN infection in Pluronic gel medium

Screening of 272 rice accessions in Petri dishes (containing Pluronic gel/PF-127 medium) revealed variability in response to RRKN infection at 16 days post inoculation (dpi). Fig 1A and 1B show the frequency distribution of the gall numbers and nematode multiplication factor (MF), respectively, while all values (including numbers of total endoparasites, egg masses and eggs per egg mass) are detailed in S3 Table. According to Fig 1A, 49 of the accessions showed highly resistant response (0–2 mean number of galls), 8 exhibited highly susceptible response (7–12 galls), while 87 accessions showed moderately resistant response (4–5 galls). The mean number of galls was recorded to be 7–8 in reference genotypes Taipei 309 and Pusa 1121. A similar trend was observed in terms of MF; a moderate MF value of 12–20 was calculated in maximum of 89 accessions (Fig 1B). The results of a repeat run with 50 randomly selected accessions revealed a strong correlation with the initial gall ($R^2 = 0.780$, $P < 0.001$) and MF ($R^2 = 0.720$, $P < 0.001$) data, indicating the high reproducibility of PF-127-based assay.

When all the genotypes were categorized into different taxonomic groups of wild rice, accessions from *O. nivara* (2.9 mean numbers of galls) showed the least number of galls as compared to the accessions that are classified as *O. rufipogon* (4.1) and *O. spontanea* (5.2). A significant difference ($P < 0.01$, $n > 30$) in gall numbers between different species (Fig 1C) was found, which explains the genetic variation between wild rice species in our study. When

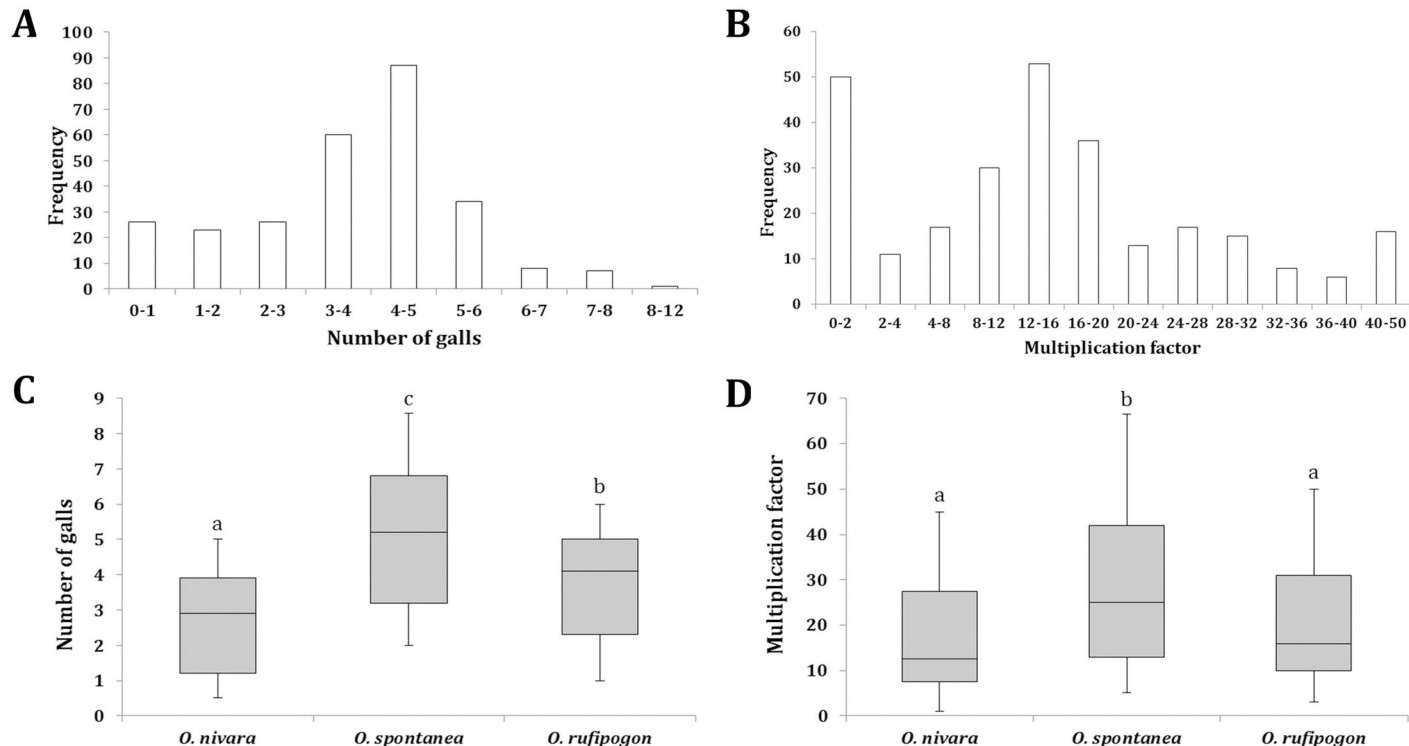

**Fig 1. Host response of 272 rice accessions to *M. graminicola* infection at 16 dpi (inoculum level– 30 J2s per plant) in PF-127 medium.** Frequency distribution of gall numbers (A) and nematode multiplication factor (B) in 272 genotypes. Box plot of relative gall numbers (C) and multiplication factor (D) across the different taxonomic groups of wild rice, namely *O. nivara*, *O. rufipogon* and *O. spontanea*. Box plots with the same letters are not significantly different ($P > 0.01$, Tukey's HSD test; $n > 30$).

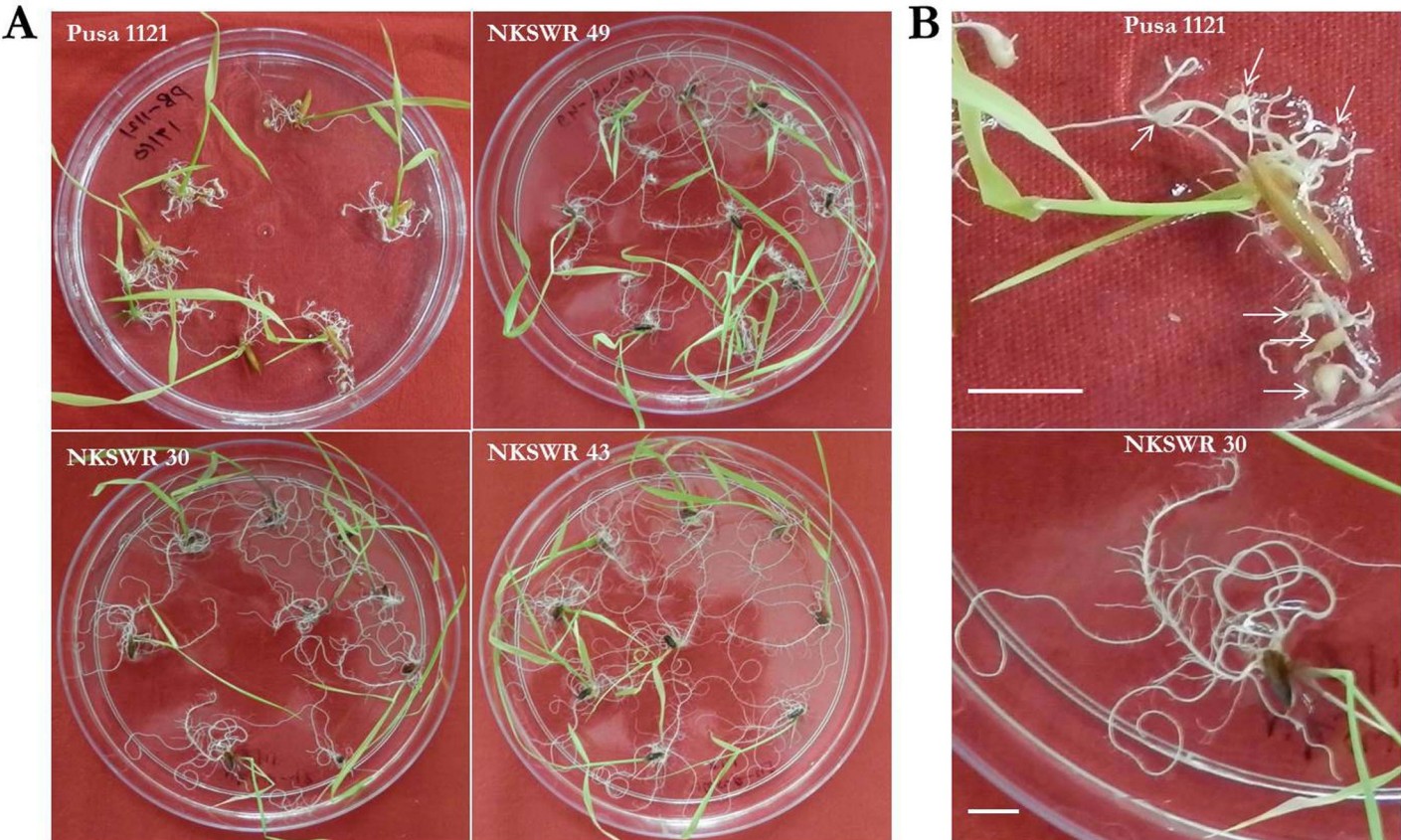

**Fig 2. RRKN infected plantlets of selective accessions in Petri dishes containing PF-127 medium at 16 dpi (inoculum level– 30 J2s per plant).** (A) The susceptible check Pusa 1121 had 7.88 ± 0.44 galls in each plantlet. A wild accession NKSWR 49 exhibited 7.25 ± 0.88 mean number of galls. By contrast, NKSWR 30 and NKSWR 43 showed zero and 0.38 ± 0.18 galls, respectively. (B) A magnified view shows typical hook-shaped galls (indicated as white arrows) in Pusa 1121 and no gall in NKSWR 30. Scale bar = 1 cm.

MF values were taken into account a significant difference ($P < 0.01$, $n > 30$) was observed between *O. spontanea* and *O. nivara*/*O. rufipogon* but not between *O. nivara* and *O. rufipogon* (Fig 1D). When accessions were stratified into different agro-climatic zones (independent of species), no significant difference was observed. This may be because each geographic location contained all the three species of wild rice (S1 Table). Although accessions from Mid-Gangetic Plains (MGP; majority of them were *O. nivara*) showed the least number of galls compared to accessions from other geographic locations (data not shown).

NKSWR 30, an *O. nivara* accession from MGP, showed no gall in any of the twelve replicates (Fig 2A and 2B; S3 Table) whereas NKSWR 11, NKSWR 13, NKSWR 15, NKSWR 18, NKSWR 19, NKSWR 23, NKSWR 25, NKSWR 43, NKSWR 48, NKSWR 108, NKSWR 123, NKSWR 124, NKSWR 128, NKSWR 141, NKSWR 144, NKSWR 156, NKSWR 160 (*O. nivara*, MGP), NKSWR 259 (*O. nivara*, Gujarat Plains and Hills (GPH)), NKSWR 101 (*O. spontanea*, MGP) and IC 336687 (*O. rufipogon*, East Plateau Hills (EPH)) had only one gall in some of the replicates and zero galls in the remaining replicates (S3 Table).

## Host response of 40 rice accessions to RRKN infection in soil

Based on extremely low counts ($< 2$) of gall numbers and MF in PF-127 assay, 40 wild rice accessions were further screened for *M. graminicola* resistance in soil. Absolute numbers of

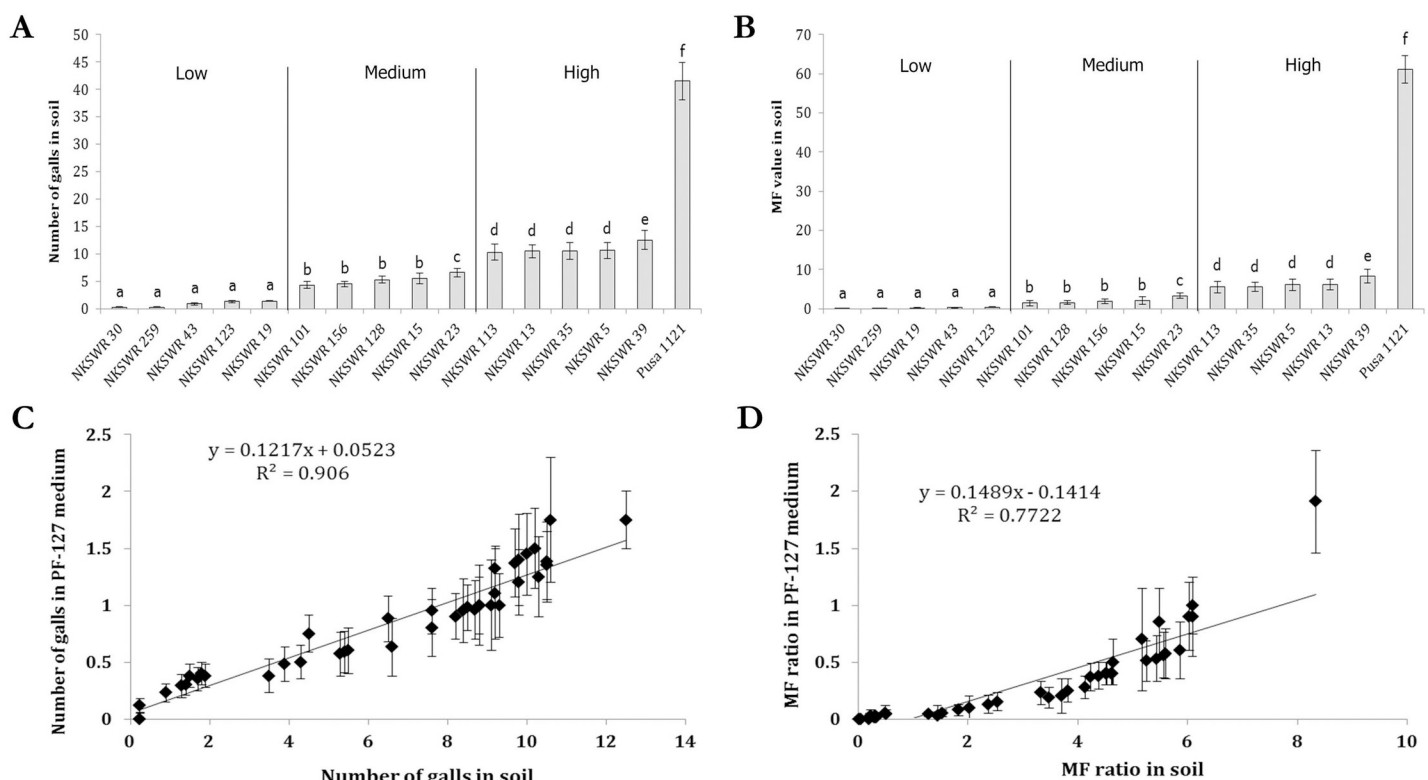

**Fig 3. Host response of selective rice accessions to *M. graminicola* infection at 16 dpi (inoculum level– 200 J2s per plant) in soil.** Accessions were grouped into five each for low, medium and high incidence of gall numbers (A) and nematode multiplication factor (B). Data of reference genotype Pusa 1121 is shown. Each bar represents the mean ± standard error (*n* = 12). Genotypes having different letters indicate significant difference at *P* < 0.01, Tukey's HSD test. Plots of mean and standard error of gall numbers (C) and MF values (D) in all the 40 accessions in soil versus the screening of identical genotypes in PF-127 medium at 16 dpi.

galls, endoparasites, egg masses, eggs per egg mass and the resultant MF ratio at 16 dpi are detailed in S4 Table. As an example, the results for some accessions that show low (NKSWR 30, NKSWR 259, NKSWR 43, NKSWR 123, NKSWR 19), medium (NKSWR 101, NKSWR 156, NKSWR 128, NKSWR 15, NKSWR 23) and high (NKSWR 113, NKSWR 13, NKSWR 35, NKSWR 5, NKSWR 39) gall numbers and MF are presented in Fig 3A and 3B. However, compared to the reference genotype Pusa 1121 (gall number– 41.5 ± 3.44; MF– 61.14 ± 5.95) RRKN infection in the highest susceptible wild rice genotype NKSWR 39 (gall number– 12.5 ± 1.75; MF– 8.33 ± 2.25) was remarkably lower (*P* < 0.01; Fig 3A and 3B). In accordance with the PF-127 assay, genotypes NKSWR 30 (gall number– 0.25 ± 0.05; MF– 0.02 ± 0.01) and NKSWR 259 (gall number– 0.25 ± 0.1; MF– 0.05 ± 0.03) showed highly resistant response indicating no J2s had penetrated the roots in the majority of the replicates. Moreover, with reference to Taipei 309 and Pusa 1121, comparatively lower number of egg masses, eggs per egg mass and the resultant MF ratio in all the 40 genotypes (S4 Table) indicate the retarded nematode development and reproduction in selected wild rice accessions. A similar trend was observed when a highly stringent assay was conducted with 40 accessions in PF-127 medium (data not shown).

The number of galls and MF ratio documented in all the 40 accessions in soil assay had correlated very strongly with the identical accessions in PF-127 assay on same parameters ($R^2$ = 0.906 (gall number), $R^2$ = 0.772 (MF), *P* < 0.001; Fig 3C and 3D).

## Population structure and genetic diversity analysis of wild rice accessions

For determining the population structure of 270 wild rice accessions and one each of *japonica* (Taipei 309) and *indica* (Pusa 1121) cultivars, genotyping was performed using genome-wide 50K SNPs with no missing data. The Bayesian model-based analysis of population structure of 272 accessions showed the optimum population structure at K = 3 (Fig 4A), suggesting these accessions were stratified into three major genetic sub-populations. Population I (indicated in red color) consisted 38.60% of accessions (105) in which 84 accessions were pure and 21 were admixed. Population II (indicated in green color) accounted for 22.04% of accessions (60) of which 17 were admixed. Population III (indicated in blue color) accounted for 39.33% of accessions (107) of which 19 were admixed (Fig 4B; S3 Fig). The mean Fst values for population I, II, and III were 0.4691, 0.6877 and 0.8366, respectively, exemplifying a considerable proportion of admixture types in our wild rice accessions possibly because of natural intercrossing among these sub-populations. Based on our earlier studies with subsets of these wild rice accessions and their co-clustering with known *Aus* and *Indica* rice cultivars ([24]; 418 wild rice accessions were genotyped via 48-plex Illumina GoldenGate assay), the populations I and II are designated as 'Pro-Aus' and 'Pro-Indica' populations, respectively. Pro-Aus and Pro-Indica populations were distributed throughout India, whereas population III was primarily concentrated in MGP and thus designated as 'Mid-Gangetic' population (Fig 4B; S1 Table). Notably, Pro-Indica populations contained the highest percentage (28.33%) of admixture types compared to Pro-Aus (20%) and Mid-Gangetic (17.75%) populations. The results of principal

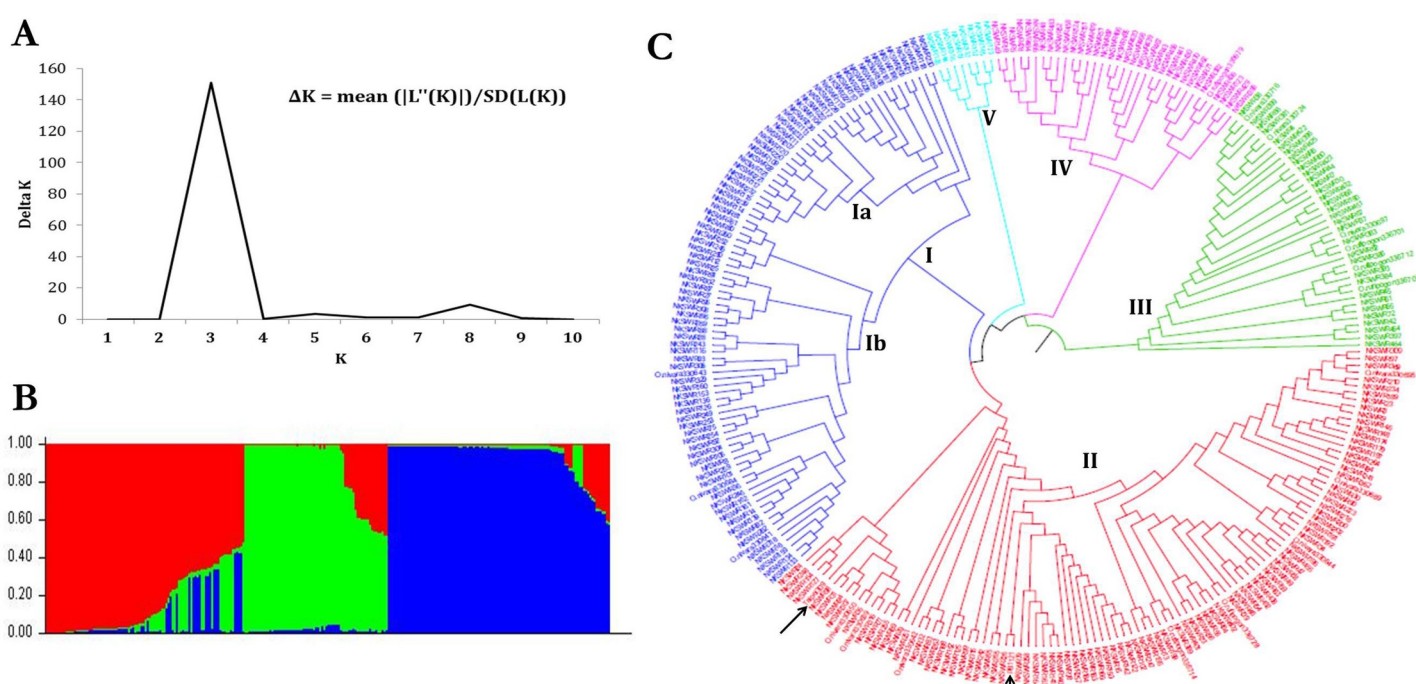

**Fig 4. Population structure and genetic diversity of 272 rice genotypes based on genome-wide 50K SNP chip.** (A) Estimation of the number of populations (K) in wild rice accessions using ΔK values. Three subgroups inferred from STRUCTURE analysis. (B) Distribution of rice accessions into three different populations. The colored region grouped the accessions in corresponding populations as red (Pro-Aus), green (Pro-Indica) and blue (Mid-Gangetic). *X*- and *Y*-axis indicate rice accessions and % membership to a genetic group, respectively. (C) SNP haplotype-based neighbor-joining phylogenetic tree of 272 accessions shows wide genetic diversity among wild rice accessions. Five different clusters (highlighted as blue, red, green, magenta and sky blue) contained nine distinct sub-clusters. Reference genotypes (Taipei 309 and Pusa 1121) are indicated by arrows.

component and kinship matrix analysis were consistent with the finding that our association panel consisted of three different groups of population (S4 Fig).

Diversity analysis revealed 272 rice accessions were grouped into five major clusters (Fig 4C). Cluster I was split into two sub-clusters, in which majority of Mid-Gangetic populations were nested into sub-cluster Ia and sub-cluster Ib was of mixed type. Majority of Pro-Aus populations were grouped into cluster II which was further subdivided into three sub-clusters each containing a mixture of Pro-Indica and Mid-Gangetic populations. Cluster III and IV was of admixed type containing accessions from diverse agro-climatic zones such as MGP, Upper Gangetic Plains (UGP), Lower Gangetic Plains (LGP), GPH, West Himalayan Regions (WHR), East Himalayan Regions (HER), East Coastal Plains (ECP) and Island Regency (IR). Cluster V nested *O. nivara* accessions from MGP (Fig 4C). Our results suggested that population structure analysis may not always conform with diversity analysis as in former case inherent genetic structure are inferred based on their ancestral lineage while in latter genotypes can be grouped into additional clusters according to their geographical origin.

## Genome-wide association analysis (GWAS)

The filtered SNPs detected via Axiom Analysis Suite (version 2.0) were used for GWAS after incorporating the phenotypic data. These SNPs provide a whole genome-wide coverage along the 12 chromosomes of rice (Fig 5). A total of 17 significant SNPs associated with *M. graminicola* resistance (based on four different parameters including gall number, egg mass, eggs per egg mass and MF) to rice were identified by GWAS using mixed linear model (MLM), which controlled for population structure and familial relatedness (Fig 6A and 6B; Table 1). The 17 association SNPs explained 5.33–12.61% of the total phenotypic variation (Table 1). We identified two SNPs to be associated with gall number, appearing to represent two QTLs on chromosome 2 and 6 in rice (threshold for significance–$\log_{10}(P) > 4$; Fig 6; Table 1). Two SNPs each were associated with egg mass (on chromosome 2 and 4) and eggs per egg mass (in chromosome 1 and 11). No significant SNPs for total endoparasites were observed (only one SNP was detected at–$\log_{10}(P) > 2.75$). The greatest number of SNPs (9) was detected (in chromosome 1, 2, 3, 4, 6, 10 and 11) to be associated with the most important trait, i.e. nematode multiplication factor (–$\log_{10}(P) > 4$; Fig 6; Table 1). Overall, two QTLs on chromosome 1 and 4 were found to be associated with all the traits (Table 1). Percent heritability explained by corresponding SNPs for phenotypic traits such as gall number, egg mass, eggs per egg mass and MF were 78, 71, 65 and 89%, respectively (Table 2). The relationship between 40 resistant accessions and genotypes of significant SNPs are described in S5 Fig. Notably, polymorphism occurred in four different allelic combinations (TC, AG, CG and AT) across the genotypes. Accessions NKSWR101 (*O. spontanea*), NKSWR259, NKSWR30 (*O. nivara*), and IC336687 (*O. rufipogon*) which earlier conferred greatest degree of resistance to *M. graminicola* also harbored greatest number of SNPs.

In accordance with Dimkpa et al. [23], herein, QTLs were considered to be within 200 kb of a significant SNP. This 200 kb window falls within the estimated window of LD decay (~50–500 kb) in rice [35–38]. More specifically, in wild rice LD decays at approximately 1–200 kb (*O. nivara* maintains LD over larger distance due to their high level of self-pollination; [39]). Notably, *O. nivara* constituted 66.67% of total wild rice accessions in our study (S1 Table). Our genotyping data of 272 accessions revealed that the average LD decayed at 218 Kb, with whole panel $r^2$ estimate of 0.2 for the 50K chip and the highest $r^2$ estimate was 0.4. The overall LD decay was slower (Fig 7). Based on the detailed annotations in Rice Genome Annotation Project and NCBI, we identified candidate genes either flanking or within the 200 kb region on each side of the 17 associated SNPs (Table 1). A number of transcription factors (TFs)

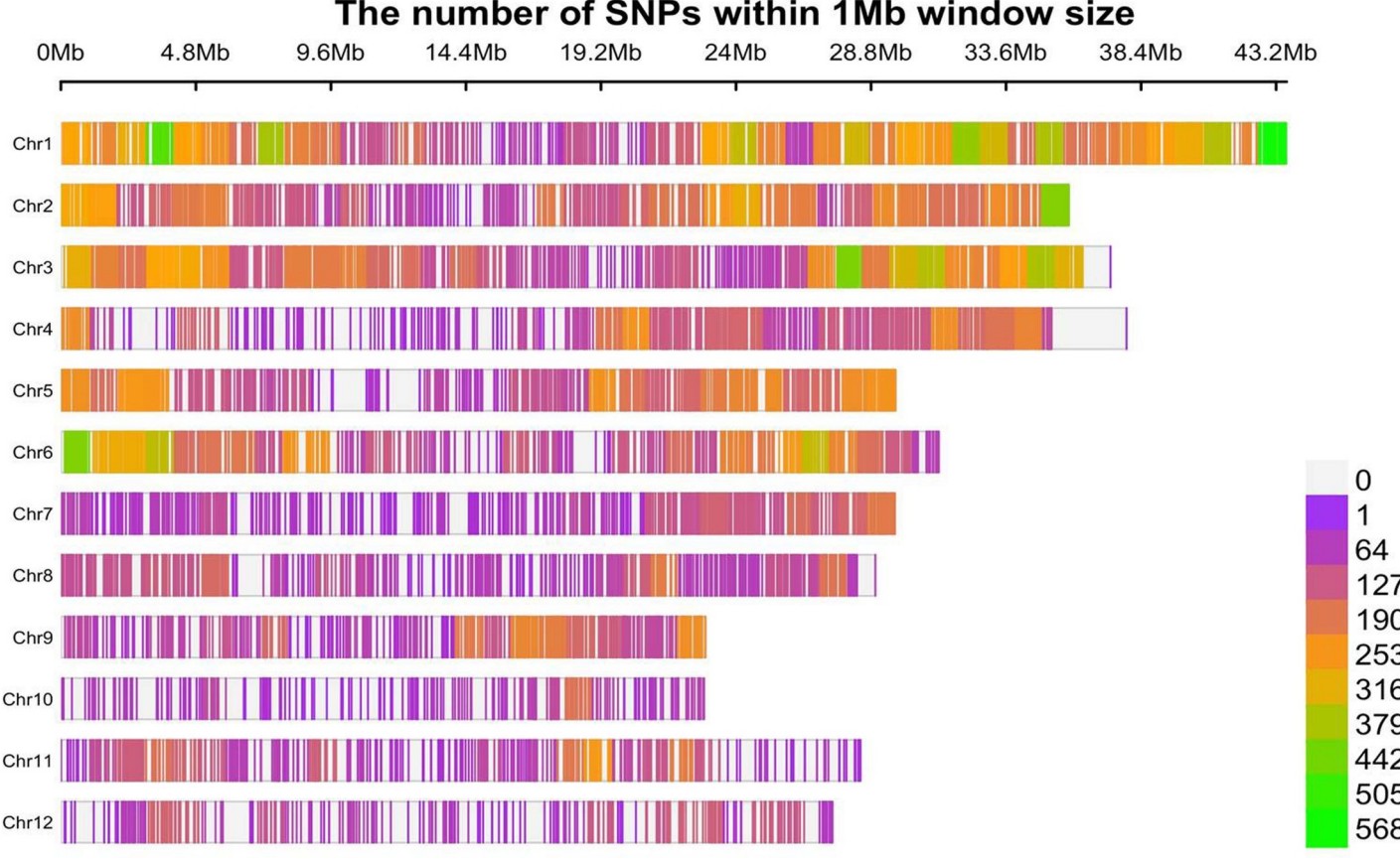

**Fig 5. SNP density plot across the 12 chromosomes of rice representing number of SNPs within 1 Mb window size.** The horizontal axis represents the chromosome length in Mb. Different colors correspond to SNP density.

including leucine zipper (bZIP), SCARECROW, MYB, MADS-box, WRKY, auxin response factor (ARF), GRAS etc. (regulate plant defense gene expression during pathogen stress) were located on different chromosomes (Table 1). Several resistance proteins such as leucine-rich repeat (LRR), NBS-LRR, Cf2/Cf5, RGA3, Lr34 analog ABC transporter, zinc finger motifs, WD repeat, F-box (analogous to WD and LRR) were located in different QTLs (Table 1). Additionally, a number of protein kinases, Exo70 exocyst complex and kelch motifs were identified which are not listed in Table 1. Gibberellin, pathogenesis-related (PR) proteins, HEAT repeats and 14-3-3 proteins which are required for plant immunity to pathogen stress were also identified. Aquaporin, nodulin and auxin regulatory proteins were identified which are known to regulate the giant cell initiation and maintenance during nematode infection (Table 1).

Interestingly, a number of NBS-LRR proteins were located in QTL 11.2. Similarly, QTLs 1.3 and 4.2 were rich in WRKY TFs and zinc fingers, respectively. However, many candidate genes were common between QTL 2.2 and 2.3 ($r^2 = 0.75$), 4.2 and 4.3 ($r^2 = 0.7$), 4.3 and 4.4 ($r^2 = 0.55$), as they were within 200-kb LD (Table 1). SNPs were located within the candidate genes in QTL 1.3 (ubiquitin-conjugating receptor kinase), 2.1 (ADP-ribosylation factor), 2.3 (DEAD/DEAH-box helicase), 4.2 (zinc finger) and 6.2 (GTP-binding Rac protein).

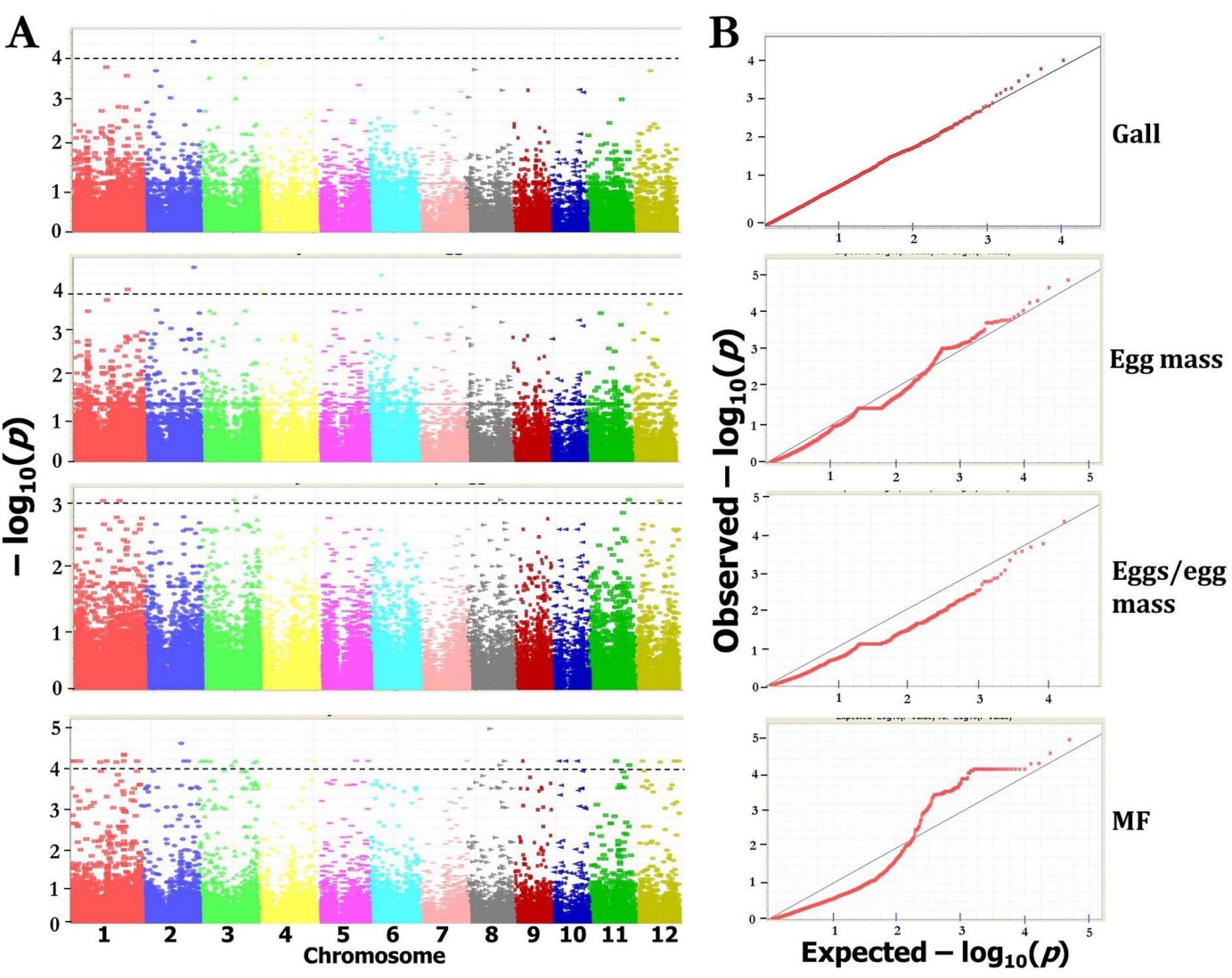

**Fig 6. Genome-wide association analyses for RRKN resistance in wild rice.** (A) Manhattan plots of *P*-values from the mixed linear model (MLM) for different traits such as numbers of galls, egg masses, eggs per egg mass and MF value. *X*-axis indicates the SNP location along the 12 chromosomes, *Y*-axis is the–log$_{10}$ (*P*-value). The dashed line represents the threshold defined by 1000 permutations of the association analysis. (B) Quantile-Quantile plots for different phenotypic traits with mixed model results. Dotted red line is the one-to-one line. *X*-axis represents expected–log$_{10}$ (*p*) and *Y*-axis is observed–log$_{10}$ (*p*) of each SNPs.

## RT-qPCR-based expression analysis of candidate genes

To further test the role of selective candidate genes in resistance of rice to *M. graminicola* infection, RT-qPCR was performed in infected and uninfected root tissues of Pusa 1121 seedlings. Expression of the fourteen genes was quantified in whole roots collected at the time of nematode inoculation and at 7 dpi in infected and uninfected plants. This enabled us to investigate the developmental regulation of the genes in young rice seedlings, as well as their regulation upon nematode infection. Except LOC_Os01g18440 (MADS-box), Os02g22140 (ADP-ribosylating protein), Os03g48450 (SCARECROW TF), Os04g50070 (zinc finger) and Os11g34460 (F-box), all the candidate genes were significantly (*P* < 0.01) upregulated in mock-inoculated plants (7 dpi) compared to 0 dpi control plants (Fig 8). The temporal expression of Os01g11200 (MYB TF), ADP-ribosylating protein, SCARECROW TF, zinc finger, Os06g47260 (Rac protein), Os11g05640 (bZIP TF), Os11g10720 (Cf2/Cf5), Os11g10760

**Table 1. Significant SNP markers associated with QTLs conferring resistance to *M. graminicola* in 272 rice accessions.**

| Suspected QTLs | Trait | SNP ID | Chromosome | Position (bp) | *P*-value | $R^2$ (%) | MAF | SNP located in gene | Genes in 200 kb LD |
|---|---|---|---|---|---|---|---|---|---|
| 1.1 | Eggs/egg mass | AX-95940548 | 1 | 6,008,298 | 5.82E-04 | 12.61 | 0.49 | LOC_Os01g11230 (heme-binding protein) | Os01g11110, 11350 (bZIP TF), Os01g11200 (MYB TF), Os01g11150 (gibberellin) |
| 1.2 | MF | AX-95961959 | 1 | 10,396,769 | 4.30E-04 | 7.72 | 0.52 | Os01g18490 (expressed protein) | Os01g18440 (MADS-box protein), Os01g18584 (WRKY9) |
| 1.3 | Galls, Egg mass, Eggs/egg mass, MF | AX-95964807 | 1 | 34,895,044 | 3.24E-04 | 7.49 | 0.44 | Os01g60330 (ubiquitin-conjugating receptor kinase) | Os01g60060 (leucine-rich repeat), Os01g60440 (HEAT repeat), Os01g60490 (WRKY22), Os01g60520 (WRKY116), Os01g60540 (WRKY20), Os01g60600 (WRKY108) |
| 2.1 | MF | AX-95945545 | 2 | 13,183,197 | 9.42E-04 | 6.69 | 0.39 | Os02g22140 (GTP binding ADP-ribosylation factor) | Os02g22020 (MYB TF) |
| 2.2 | Galls | AX-95921578 | 2 | 24,480,888 | 2.47E-04 | 9.24 | 0.32 | Os02g40420 (expressed protein) | Os02g40430 (HEAT repeat), Os02g40450 (DEAD/DEAH box), Os02g40530 (MYB TF), Os02g40664 (zinc finger) |
| 2.3 | Egg mass | AX-95947392 | 2 | 24,526,132 | 8.59E-04 | 8.32 | 0.33 | Os02g40450 (DEAD/DEAH box) | Same as above |
| 3.1 | MF | AX-95935442 | 3 | 27,655,564 | 8.69E-04 | 10.35 | 0.30 | Os03g48490 (centromere protein) | Os03g48450 (SCARECROW TF), Os03g48600 (auxin-regulated protein, DUF966) |
| 4.1 | Egg mass | AX-95925342 | 4 | 27,068,142 | 1.22E-05 | 10.89 | 0.40 | Os04g45740 (hypothetical protein) | Os04g45460 (cysteine-rich secretory protein), Os04g45690 (B-box zinc finger), Os04g45810 (leucine zipper), Os04g45960 (subtilisin protease), Os04g46020 (GATA zinc finger), Os04g46060 (WRKY36) |
| 4.2 | MF | AX-95964658 | 4 | 29,863,576 | 4.84E-05 | 7.92 | 0.28 | Os04g50070 (C2H2 zinc finger) | Os04g49890, Os04g49900 (ABC transporter), Os04g49950 (F-box protein), Os04g49980 (LEA protein), Os04g50060 (GRAS TF), Os04g50100 (RING H2 zinc finger), Os04g50200 (F-box protein), Os04g50660 (WD protein), Os04g50680, 50770 (MYB TF), Os04g50700, 50710 (PR protein), Os04g50720, 50740, 50760 (RING zinc finger) |
| 4.3 | MF | AX-95924928 | 4 | 29,916,160 | 9.55E-04 | 6.43 | 0.39 | Os04g50160 (retrotransposon protein) | Os04g50660 (WD protein), Os04g50680, 50770 (MYB TF), Os04g50700, 50710 (PR protein), Os04g50720, 50740, 50760 (RING zinc finger), Os04g50920 (WRKY37) |
| 4.4 | Galls, Egg mass, Eggs/egg mass, MF | AX-95939714 | 4 | 30,155,892 | 4.56E-04 | 9.89 | 0.42 | NA | Same as above, Os04g51172 (zinc finger) |
| 6.1 | MF | AX-95936929 | 6 | 28,018,736 | 4.21E-05 | 7.32 | 0.35 | Os06g46250 (expressed protein) | Os06g45970 (auxin-responsive gene), Os06g46240 (BTB/POZ domain), Os06g46366 (C3HC4 zinc finger), Os06g46410 (auxin response factor), |
| 6.2 | Galls | AX-95927509 | 6 | 28,651,370 | 9.18E-06 | 9.91 | 0.38 | Os06g47260 (GTP binding Rac protein) | Os06g47150 (auxin response factor), Os06g47200 (protease inhibitor), Os06g47230 (coiled-coil domain), Os06g47270 (C3HC4 zinc finger) |
| 10.1 | MF | AX-95938411 | 10 | 17,990,076 | 8.05E-05 | 8.53 | 0.28 | NA | Os10g33810 (MYB TF), Os10g33940 (auxin response factor), Os10g34000 (aquaporin), Os10g34030 (C3HC4 zinc finger), Os34040 (nodulin) |
| 11.1 | MF | AX-95961774 | 11 | 2,694,492 | 2.16E-05 | 5.33 | 0.36 | NA | Os11g05640 (bZIP TF), Os11g05660 (F-box protein), Os11g05800 (abscisic acid induced HVA22) |

(*Continued*)

**Table 1.** (Continued)

| Suspected QTLs | Trait | SNP ID | Chromosome | Position (bp) | P-value | $R^2$ (%) | MAF | SNP located in gene | Genes in 200 kb LD |
|---|---|---|---|---|---|---|---|---|---|
| 11.2 | MF | AX-95933197 | 11 | 5,848,427 | 8.98E-04 | 7.13 | 0.37 | NA | Os11g10550, 10570, 10610, 10620, 10760 (NBS-LRR protein), Os11g10720 (cf2/cf5 resistance gene), Os11g10770 (RGA3 resistance protein) |
| 11.3 | Eggs/egg mass | AX-95960766 | 11 | 20,145,250 | 4.00E-04 | 10.75 | 0.35 | Os11g34370 (phospholipase) | Os11g34450 (14-3-3 protein), Os11g34460 (F-box protein), Os11g34660 (protease inhibitor), Os11g34700 (C2H2 zinc finger) |

Note that QTL 1.3 and 4.4 are common for all the traits. $R^2$ –phenotypic variation explained by the corresponding SNP. Rice locus IDs are provided as gene models. TF–transcription factor.

(NBS-LRR) and Os11g34450 (14-3-3) was significantly ($P < 0.05$) elevated in RRKN-infected roots at 7 days after inoculation (Fig 8), suggesting these genes may have a role in eliciting the defense response against *M. graminicola*. In contrast, expression of MADS-box, Os02g40450 (DEAD-box), Os04g50660 (WD repeat) and F-box protein was unaltered ($P > 0.01$) in 7 dpi infected plants compared to 7 dpi uninfected plants. Intriguingly, steady-state mRNA levels of Os10g33940 (ARF18) was significantly ($P < 0.01$) attenuated in 7 dpi infected plants than 7 dpi uninfected ones (Fig 8), indicating the role of this gene in rice to RRKN susceptibility.

## Discussion

### Broad and novel genetic resources for *M. graminicola* resistance in rice

The wild relatives of rice harbor rich and novel genetic resources which can be used to improve pest and disease resistance in cultivated rice [40, 41]. For example, resistance to grassy stunt virus, bacterial leaf blight, neck blast and brown plant hopper was successfully introgressed into the cultivated rice from their wild relatives [42–44]. A number of beneficial traits including tolerance to biotic and abiotic stresses have been lost in cultivated rice which possesses a narrow genetic base because of domestication and breeding bottlenecks [24, 25]. According to an estimate, modern rice varieties have retained only 20% of the genetic diversity present in their wild relatives [45]. India has unprecedented diversity of wild rice germplasm and landraces that are spread over fifteen diverse agro-climatic zones [24, 25]. Therefore, these untapped genetic resources were taken as the prime candidates for the association panel in our study in order to unravel the conserved loci that govern *M. graminicola* resistance in rice. Indeed, a large proportion (14.8%; 40 of 270 accessions) of the wild accessions was found to be highly resistant to *M. graminicola* infection in the present study.

Herein, we screened 272 diverse rice accessions collected from ten agro-climatic zones for *M. graminicola* resistance by measuring the relative numbers of galls, endoparasites, egg mass, eggs per egg mass and MF ratio in PF-127 medium. A significant variation in the susceptibility

**Table 2. Percent heritability of different phenotypic traits with corresponding number of significantly associated SNPs.** $R^2$ reflects the amount of heritable variation explained by an additive model of the SNPs (ANOVA).

| Trait | $-\text{Log}_{10}(P)$ | Significant SNPs | $R^2$ | F value | p value | % heritability |
|---|---|---|---|---|---|---|
| Galls | > 4 | 4 | 0.403169 | 12.98 | 0.0009 | 78% |
| Egg mass | > 4 | 4 | 0.363193 | 10.47 | 0.001 | 71% |
| Eggs/Egg mass | > 3 | 4 | 0.282387 | 9.78 | 0.002 | 65% |
| MF | > 4 | 11 | 0.49924 | 15.98 | 0.0002 | 89% |

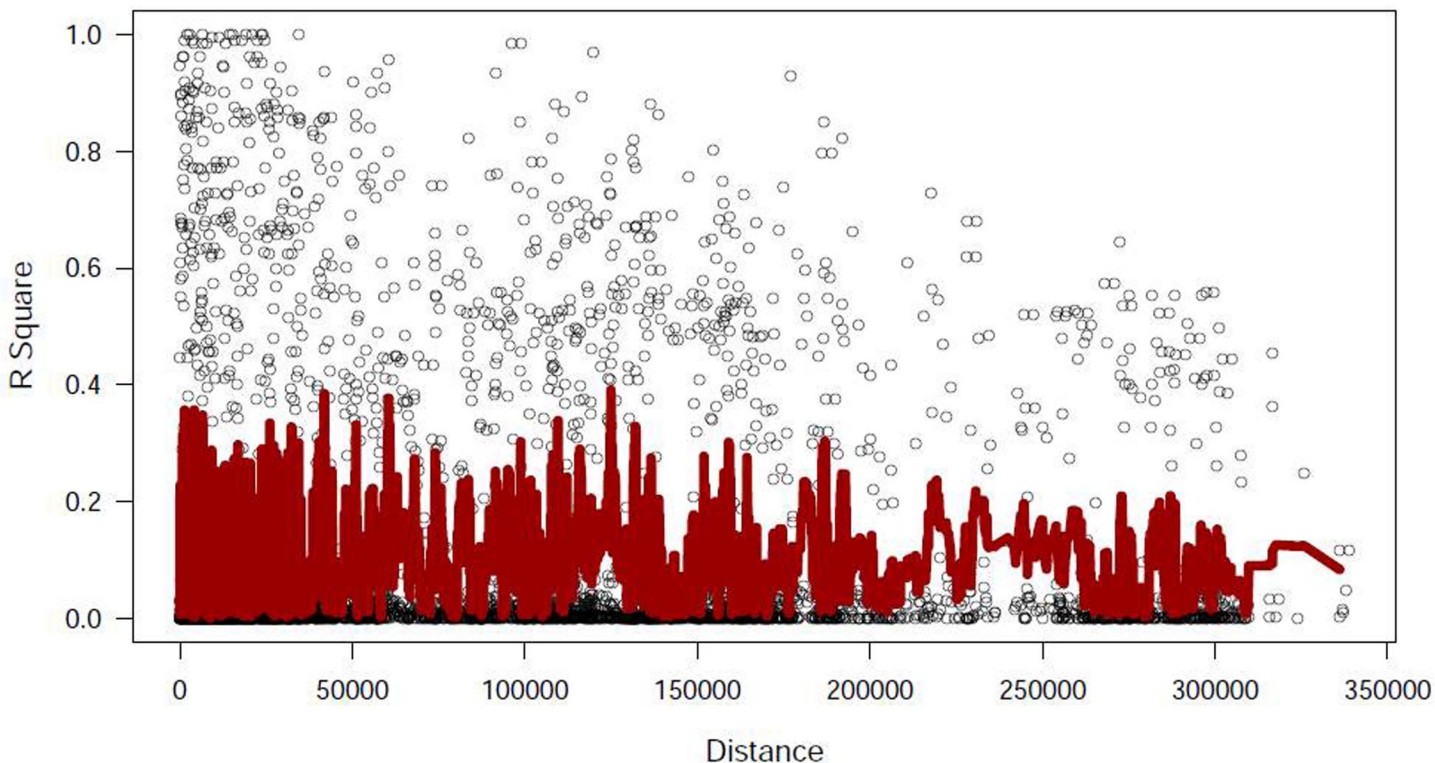

**Fig 7. Linkage disequilibrium (LD) measured $r^2$ plotted versus the physical map (bp) of 50K SNP markers in a panel of 272 genotypes.**

level of rice accessions to RRKN infection was documented. A repeat experiment with fifty randomly selected accessions from this diversity panel showed strong correlation with the initial screening results. This suggests that PF-127-based screening is a robust and reproducible method for dissecting the rice-RRKN interaction, in agreement with the previous reports from our laboratory [26, 27]. In addition, 40 highly resistant accessions were further evaluated for *M. graminicola* resistance in large plug trays containing soil as the medium. Taken together the data of both soil- and PF-127-based screening, *O. nivara* accessions NKSWR 30 and NKSWR 259 showed almost immune response to *M. graminicola* with zero galls and zero MF in majority of the replicates at 16 dpi. Other *O. nivara* accessions such as NKSWR 43, NKSWR 123, NKSWR 19, NKSWR 108, NKSWR 25, NKSWR 18 and an *O. rufipogon* accession IC 336687 also supported extremely low population level of *M. graminicola*. The negligible susceptibility of these accessions cannot be attributed to poor root growth because root weight of these accessions was comparable with that of susceptible accessions (data not shown). The differential susceptibility of 332 *O. sativa* accessions to *M. graminicola* was earlier reported via soil-based screening in which two accessions LD 24 (*indica*) and Khao Pahk Maw (*aus*) were found to be almost immune to RRKN infection [23, 46].

Among 40 highly resistant wild accessions, 34 belonged to *O. nivara* type (33 of them collected from middle Gangetic plains agro-climatic zone) whereas 3 each belonged to *O. rufipogon* and *O. sativa* f. *spontanea* types. According to PF-127-based screening, a clear difference in resistance level in 272 accessions was also evident when genotypes were categorized into different taxonomic groups, i.e. *nivara*-, *rufipogon*- and *spontanea*-type. These differences can be explained by the possibility of different selection pressures among the geographic regions. When compared within different agro-climatic zones, accessions from MGP showed least

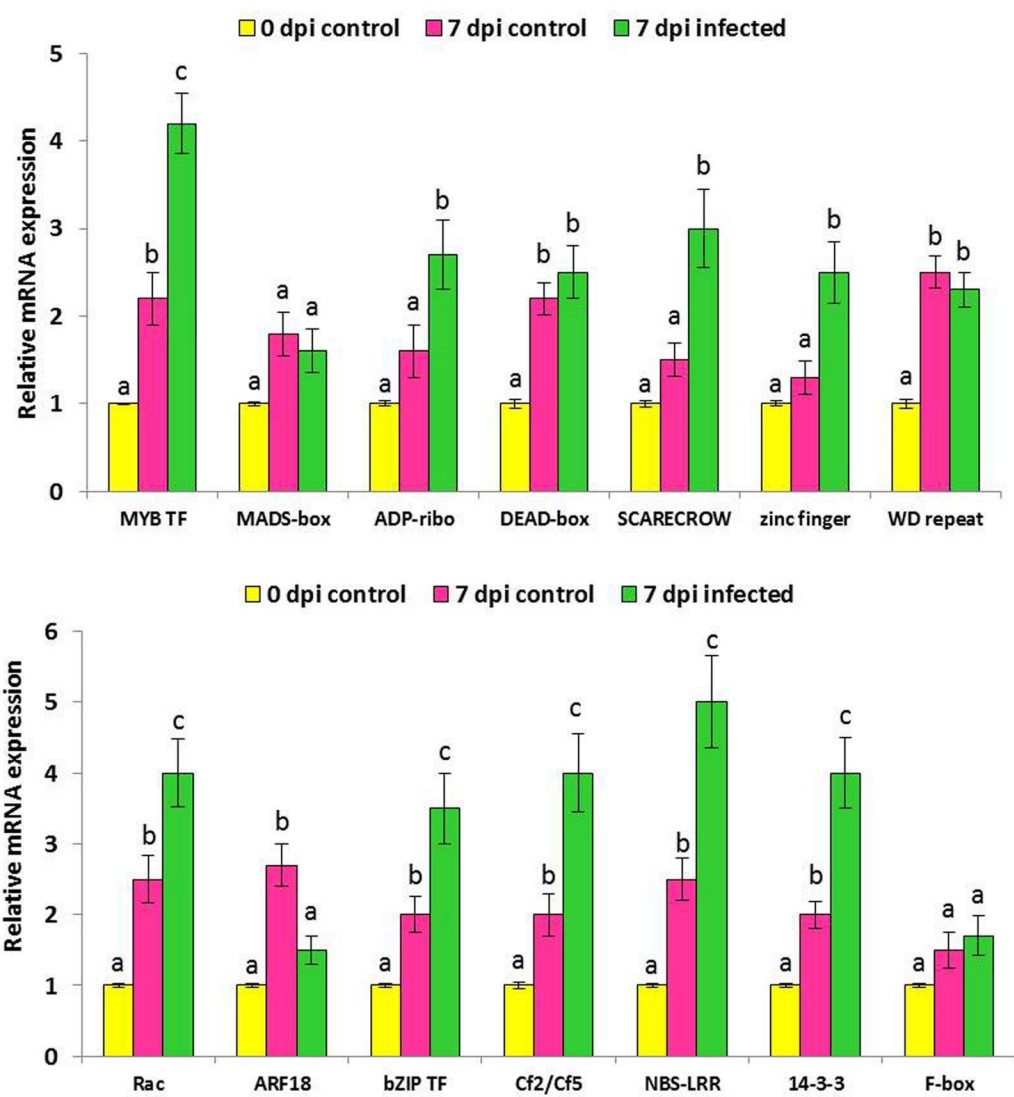

**Fig 8. RT-qPCR-based differential expression patterns of 14 candidate genes in infected and uninfected root tissues of Pusa 1121 plants at 7 days post inoculation (dpi) with *M. graminicola*.** Pusa 1121 was used as the reference cultivar. Data reflect gene expression levels in whole roots collected at the time of nematode inoculation (0 dpi control), in whole roots at 7 days after mock inoculation (7 dpi control) and at 7 days after nematode inoculation (7 dpi infected). Data are expressed as the normalized fold change in expression relative to the internal reference gene of *O. sativa*, i.e. actin. Bars represent mean expression levels ± standard error from three biological and three technical replicates each containing a pool of ten plants. Different lower-case letters (within each gene) indicate statistical difference determined by ANOVA with post-hoc Tukey's HSD test, $P < 0.01$.

nematode infection. The majority of resistant accessions originally from middle Gangetic plains (occupies the eastern part of Uttar Pradesh and Bihar) presumably have experienced higher selection pressure from RRKN infection. A similar stratification of cyst nematode (*Heterodera glycines*) resistance level across the different geographic location of soybean crop was reported when 235 wild accessions were included for association mapping analysis [22].

The genetic variation among Indian wild rice accessions were studied via model-based population structure analysis using genome-wide unlinked SNP markers (unlinked markers provide high reproducibility and success rate of population structure analysis; [47]) which grouped the accessions into three distinct subpopulations, namely Pro-Aus, Pro-Indica and

Mid-Gangetic populations. Additionally, genetic diversity and cluster analysis based on SNP markers revealed the current wild accessions were highly diverse indicating each populations of an agro-climatic zone constituted mixture of genetically diverse individuals. According to the Fst values, Pro-Indica and Mid-Gangetic populations contained the greatest and least proportion of admixture types, respectively. Thus, accessions collected from middle Gangetic plains agro-climatic zone represented a conserved wild rice subpopulation whereas relatively greater level of gene flow and outcrossing was speculated among the geographically adjacent than the distant populations. Notably, overlapping geographic distribution pattern of wild rice subpopulations was reported by several studies [39, 48, 49]. Due to their open inflorescence the sympatric species, *O. nivara* and *O. rufipogon* can outcross with each other and also with cultivated *O. sativa* [50, 51]. Nevertheless, the greater genetic diversity of present wild rice accessions is in line with the hypothesis that wild rice are the source of useful new genes for future varietal improvement program [29, 40].

Phenotyping of the association panel has offered important information about the degree and distribution of RRKN susceptibility in Indian wild rice species including *O. nivara*, *O. rufipogon* and *O. sativa* f. *spontanea*, and in two cultivated species, i.e. *O. sativa indica* and *japonica*. This provided new insights into the evolution of resistance/susceptibility of wild and cultivated rice species to *M. graminicola* parasitism. The higher susceptibility of *indica* and especially *japonica* rice is not surprising maybe because no selection is in function to introduce resistance in *japonica* as RRKN is not a pest of temperate rice cultivation [23]. Intriguingly, a majority of *O. nivara* accessions from middle Gangetic plains (a suspected genetic diversity hotspot; [24]) has shown the least susceptibility to RRKN infection. This may be because of their eco-geographical isolation in accordance with the rationale that gene flow is inversely related to geographic distance of natural populations [52].

## Genetic loci determining the parasitic success of *M. graminicola* in rice

By analyzing the marker-trait associations in 272 rice accessions via GWAS, we identified 17 significant SNPs that governed the RRKN resistance-related traits such as numbers of galls, egg mass, eggs per egg mass and MF ratio. By applying the $-\log_{10}(P)$ score of 4 as a threshold for significance, we identified two QTLs at chromosome 1 and 4 of rice were associated with all the traits. A similar stringency level or a less stringent threshold ($-\log_{10}(P) > 3$) was adopted while reporting significant SNPs associated with nematode resistance/susceptibility in rice [23], wheat [21], soybean [22] and Arabidopsis [53]. In particular, four SNPs in chromosome 4, three SNPs each in chromosome 1, 2, 11, two SNPs in chromosome 6, and one SNP each in chromosome 3 and 10 were found to be associated with different traits. The trait such as total endoparasite counts in rice accessions was not taken into consideration as only one SNP was detected for this trait at $-\log_{10}(P) > 2.75$. Lowering the threshold for significance in GWAS study may increase the false discovery rate which in turn reveals more common alleles with smaller effect size in test populations [20].

A number of transcription factors (bZIP, SCARECROW, MYB, MADS-box, WRKY, ARF, GRAS etc.) that regulate plant defense responses to nematode and various biotic stress [54–61], were located on different chromosomes within 200-kb LD of the suspected QTLs in this study. The 200 kb window was adopted because of the fact that LD decay occurs in rice at 50–500 kb [35–38]. Although in wild rice the decay is much higher at 1–200 kb, *O. nivara* maintains LD over larger distance due to their high level of self-pollination [39]. Notably, *O. nivara* constituted 66.67% of total wild rice accessions in our study. Other important candidate genes located in different QTLs include resistance genes such as NBS-LRR, Cf2/Cf5, RGA3, Lr34 analog ABC transporter, zinc finger, WD repeat, leucine-rich repeat, HEAT repeat etc. All

these genes putatively function to elicit plant innate immunity during pathogen invasion [62–65]. A number of candidate genes putatively involved in plant defense response to biotic stress were located within the SNPs in QTL 1.3 (ubiquitin-conjugating receptor kinase; [66, 67]), 2.1 (ADP-ribosylation factor; [68]), 2.3 (DEAD/DEAH-box; [69]), 4.2 (zinc finger; [64]) and 6.2 (GTP-binding Rac protein; [70, 71]) in our study.

We assume that chromosomes 1 (QTL 1.3 harbored a number of WRKY TF), 2, 4 (QTL 4.2 contained several zinc finger motifs), 6 and 11 (QTL 11.2 displayed several NBS-LRR resistance genes) might play a decisive role in contributing resistance to *M. graminicola* in rice because a number of significant SNP loci were identified in each of these chromosomes. A previous association mapping analysis have identified chromosomes 4 and 11 to harbor candidate genes such as lectin and homolog of stripe rust resistance protein [23]. Using RIL-based bi-parental mapping study, QTLs for RRKN resistance were identified on almost all the 12 chromosomes of rice to date [7, 9, 13, 14]. It is evident that RRKN resistance in rice is a complex trait that involves a number of genes which regulate a cascade of plant defense responses. Therefore, unravelling the conserved and novel RRKN-resistance loci from the non-domesticated wild rice population could improve our understanding of molecular mechanisms underlying the rice-RRKN interaction.

## Candidate genes involved in *M. graminicola* resistance in rice

Unlike a report of homologous stripe rust and powdery mildew resistance genes in chromosome 11 of *O. sativa* [23], no information is available about the canonical nucleotide-binding leucine-rich repeat (NB-LRR)-type genes that impart resistance against *M. graminicola* in rice. Herein, we report a number of NBS-LRR proteins (LOC_Os11g10550, 10570, 10610, 10620 and 10760) which was mapped on chromosome 11 of rice within 200-kb LD of QTL 11.2. Incidentally, $Hsa-1^{Og}$ gene that conferred resistance against the cyst nematode, *H. sacchari*, was mapped on chromosome 11 of *O. glaberrima* [72]. The nucleotide-binding state of NBS-LRR proteins regulates the activity of plant resistance (R) proteins that are involved in activation of plant innate immunity upon pathogen recognition [73]. The R gene in tomato contains Cf2/Cf5 locus (encode LRR) that confer resistance in tomato to *M. incognita* [62]. Upon pathogen invasion in host plants, the activated membrane-bound NBS-LRR immune receptors translocate to the cell nucleus and interact with specific transcription factors (although not exclusive, comprise members of ERF, bHLH, bZIP, MYB, NAC and WRKY families) which modulate the plant immunity [57]. In our qRT-PCR analysis, compared to uninfected plants the expression of NBS-LRR and Cf2/Cf5 protein was positively regulated in *M. graminicola*-infected Pusa 1121 at 7 dpi. In coherence, other defense response regulatory genes such as MYB TF, SCARECROW, zinc fingers, bZIP TF and 14-3-3 were upregulated in nematode-infected plants than the uninfected ones.

Overexpression of a rice ADP-ribosylation factor induced pathogen resistance in tobacco by regulating the transcript accumulation of pathogenesis-related (PR) genes and salicylic acid (SA) [68]. OsRac1 (GTP binding Rac protein) was shown to be a component of disease resistance pathway acting downstream of R gene when *OsRac1* transformed japonica rice was infected with the blast fungus, *Magnaporthe oryzae* [70]. Increased resistance to tobacco mosaic virus was documented when tobacco was transformed with *OsRac1* [71, 74]. In our qRT-PCR study, both ADP-ribosylating and Rac protein was substantially upregulated in *M. graminicola* infected plants than the uninfected ones at 7 dpi. Together, our data suggest changes in expression level of these candidate genes can be a contributory factor to confer resistance in rice to *M. graminicola*. However, expression of ARF18 TF (auxin response factor) was downregulated in nematode-infected plants compared to uninfected plants at 7 dpi in our

qRT-PCR study. Auxin-regulated proteins are known to coordinate the balance between plant root growth and disease resistance by promoting the auxin biosynthesis and suppressing the benzoxazinoid-based defense compound formation [75]. The direct role of auxin influx (AUX1, LAX3) and efflux (PIN3) proteins in giant cell formation were unraveled during Arabidopsis-*M. incognita* compatible interaction [76].

In conclusion, forty accessions displaying a high degree of resistance to *M. graminicola* were identified. This data uncovers the potential of *O. nivara*, *O. rufipogon* and *O. sativa* f. *spontanea* as novel resources for RRKN management. GWAS was successfully applied to dissect the genetic architecture of resistance to RRKN in Indian wild rice populations. The present study provides an example of exploring the untapped genetic resources and novel genes which enrich the repository of candidate genes utilizable for future marker-assisted rice breeding program for RRKN resistance.

## Supporting information

**S1 Fig. RNA extracted from the root, stem, leaf and seed tissue of *O. sativa* genotype Pusa 1121 resolved on 2% (w/v) high resolution agarose gel.** M– 1 Kb molecular weight marker.
(TIF)

**S2 Fig. Expression pattern of *O. sativa* reference gene (actin) in different plant tissues of rice genotypes (Pusa 1121 and NKSWR30) at different experimental conditions (mock and nematode inoculated).** Plants were harvested at 7 days after nematode inoculation. Pusa 1121 and NKSWR30 are susceptible and resistant to *M. graminicola* infection, respectively. Plots in bottom panel represent the quantification and melting curve of qPCR reaction.
(TIF)

**S3 Fig. Detailed graphical presentation of population structure of 272 diverse rice genotypes.** The colored region grouped the genotypes in corresponding populations as red (Pro-Aus), green (Pro-Indica) and blue (Mid-Gangetic). *X*- and *Y*-axis indicate rice accessions and % membership to a genetic group, respectively.
(TIF)

**S4 Fig.** (A) Three-dimensional plot of the principal component analysis of 272 rice genotypes. (B) Hierarchical clustering and heat map of the pairwise kinship of 272 genotypes. The color histogram depicts the distribution of coefficient of co-ancestry; stronger red color indicates more relatedness among individuals.
(TIF)

**S5 Fig. Genotypes of significant SNPs detected across the 40 resistant accessions, Pusa 1121 and Taipei 309 are the reference genotype.**
(XLSX)

**S1 Table. List of 272 rice genotypes used in the present study including wild rice or landraces (prefixed as NKSWR/IC) or improved cultivars collected from the different agro-climatic zones of India.**
(PDF)

**S2 Table. List of oligonucleotides used for qRT-PCR analysis.** Primer $T_a$– 60˚C. RAP-DB: Rice Annotation Project Database (http://rapdb.dna.affrc.go.jp/). RGAP: Rice Genome Annotation Project (http://rice.plantbiology.msu.edu/).
(PDF)

**S3 Table. Initial screening of 272 rice accessions for *M. graminicola* resistance in Petri plates containing PF-127 medium at 16 dpi.** Absolute values represented are the mean of 12 replicates. Forty wild rice genotypes (highlighted in bold letters) that exhibited the lowest number of galls and MF were selected for further studies. Reference genotypes (in bold letters) used was Pusa 1121 and Taipei 309. Most susceptible accessions are highlighted in red color.
(PDF)

**S4 Table. Screening of 40 rice accessions for *M. graminicola* resistance in pots containing soil at 16 dpi.** Absolute values represented are the mean of 12 replicates ± standard error. The reference genotypes used were Pusa 1121 and Taipei 309.
(PDF)

## Author Contributions

**Conceptualization:** Tushar K. Dutta, Uma Rao.

**Data curation:** Tushar K. Dutta, Nisha Singh.

**Formal analysis:** Tushar K. Dutta, Nisha Singh.

**Funding acquisition:** Uma Rao.

**Investigation:** Alkesh Hada.

**Methodology:** Alkesh Hada.

**Resources:** Balwant Singh, Vandna Rai, Nagendra K. Singh.

**Supervision:** Uma Rao.

**Writing – original draft:** Tushar K. Dutta.

**Writing – review & editing:** Tushar K. Dutta.

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
