## [Decision Letter · Decision Letter 0]

5 Aug 2020

PONE-D-20-22147

A genome-wide association study in Indian wild rice accessions for resistance to the root-knot nematode Meloidogyne graminicola

PLOS ONE

Dear Dr. Uma Rao,

Thank you for submitting your manuscript to PLOS ONE. After careful consideration, we feel that it has merit but does not fully meet PLOS ONE’s publication criteria as it currently stands. Therefore, we invite you to submit a revised version of the manuscript that addresses the points raised during the review process.

We look forward to receiving your revised manuscript.

Kind regards,

Ji-Zhong Wan

Academic Editor

PLOS ONE

Journal Requirements:

Additional Editor Comments (if provided):

Most comments of reviewers are positive. However, authors have to address all the concerns of reviewers, and revise the manuscript according to reviewers’ comments before acceptance.

Reviewers' comments:

Reviewer's Responses to Questions

**Comments to the Author**

1. Is the manuscript technically sound, and do the data support the conclusions?

Reviewer #1: Partly

Reviewer #2: Yes

Reviewer #3: Yes

2. Has the statistical analysis been performed appropriately and rigorously? 

Reviewer #1: Yes

Reviewer #2: Yes

Reviewer #3: Yes

3. Have the authors made all data underlying the findings in their manuscript fully available?

Reviewer #1: No

Reviewer #2: Yes

Reviewer #3: No

4. Is the manuscript presented in an intelligible fashion and written in standard English?

Reviewer #1: Yes

Reviewer #2: Yes

Reviewer #3: Yes

5. Review Comments to the Author

Reviewer #1: General comments: The paper is interesting and provides novel valuable information regarding new sources of genetic resistance in rice. My main concern regards the RT-qPCR experiments. It is not clear at all how the authors calculated the relative expression values of the selected genes. In papers reporting rigorous RT-qPCR experiments is not accepted to use only one reference gene, but at least three reference genes have to be included. I can understand that is often hard to find three or more stable reference gene, thus, in the case the authors decide to use only one reference gene, they have to demonstrate that this gene is stable under the experimental conditions. In this case the authors have to show the reaction efficiency and R2 and Cq values of the reference genes under the different experimental conditions. Moreover, the relative expression values are calculated by normalizing the Cq value of one gene to the Cq value of the reference gene and the Cq values of the mock control (and the relative expression values of the gene is calculated as follows: relative exp value= 2-ΔΔCq). In this manuscript it seems that the authors normalized the Cq values of the selected gene only to the Cq value of the reference genes, thus it is not clear at all how they calculated the relative expression value. I suggest to carefully read this paper: Bustin et al. 2009 “The MIQE guidelines: minimum information for publication of quantitative real-time PCR experiments”, Clinical Chemistry 55:4 611-622.

Please also the check the quality of your figures, some graphs are not clear, difficult to read.

Please change qRT-PCR into RT-qPCR

Line 219 “RNA quantity and quality were assessed in Nanodrop ND-1000 spectrophotometer”. To check the RNA quality is essential to run an agarose gel to verify if the extracted RNA was degraded or not. The quantification by using the Nanodrop does not provide information about the integrity of the RNA. Please, provide the gel of RNA as supplementary materials

Line 220 Please specify the starting amount of RNA used to synthetized the cDNA

Reviewer #2: Dear colleagues,

I found that the manuscript provided important novel results for the identification of QTL associated with rice resistance to root-knot nematode M. graminicola. The authors used a genetically diverse collection of wild rice, performed appropriate phenotyping and genotyping scores, and identified candidate genes for the resistance. The authors identified 40 accessions resistant to RNM, and 17 novel MTA for phenotypic traits of RNM. The provided original data is fully available. The statistical methods were applied properly. The manuscript is written using standard English, and the text is easy to read. Therefore, I am recommending to accept the manuscript for publication in PLOS ONE.

Reviewer #3: This manuscript describes variations in the responses to rice root-knot nematode (RRKN) among wild rice accessions and dissection of genetic factors conferring RRKN resistance. The authors identified 40 resistant accessions from a diversity panel of wild rice accessions using a sophisticated screening method. Using high-density SNPs data across the genome, 17 novel marker-trait associations related to RRNK resistance were detected by genome-wide association study. Furthermore, expression analysis revealed that nine genes around significant SNPs were upregulated in RRKN-infected plants indicating candidate genes related to RRKN resistance. The study is well designed, and the paper is well organized. However, it should be described the relationships between 40 resistant accessions and genotypes of significant SNPs detected. Therefore, I recommend the authors to add genotype data of the accessions and to discuss about the contribution of detected QTLs to RRKN resistant accessions. Additionally, I found some points that needed to revise or confirm. I recommend the authors to check these points carefully.

1) Relationship between resistant accessions and genotypes of significant SNPs should be described.

2) p.8, l. 226. “Table S4” should be changed to “Table S2”.

3) p.11, l. 311-312. According to the Fig. 3, the r values in the text seem to represent r square values. It should be changed to the values of correlation coefficients.

4) p.12, l. 337-338. I am confusing because you mentioned that population I and III are designated as “Pro-Aus” and “Pro-Indica”, respectively. And population II was designated as “Mid-Gangetic”. However, in the subsequent sentences, percentages of admixture types do not match with Fig. 4b. I wonder that population I, II and III are designated as “Pro-Aus”, “Pro-Indica” and “Mid-Gangetic”, respectively.

5) p.14, l. 394. “Table S2” should be changed to “Table S1”.

6) p.17, l. 431. “Os11g34450” should be changed to “Os11g34460”.

7) p.18, l. 443. “12 candidate genes” should be changed to “14 candidate genes”.

8) p.20, l. 523-525. I don’t understand the meaning of the sentence. I recommend you to rewrite it.

9) p.22, l. 557. “Rac protein; [70]; Moeder et al. [71]” should be changed to “Rac protein; [70, 71]”.

6. PLOS authors have the option to publish the peer review history of their article (what does this mean?). If published, this will include your full peer review and any attached files.

Reviewer #1: No

Reviewer #2: No

Reviewer #3: No

---

## [Author Response · Author response to Decision Letter 0]

16 Aug 2020

Response to Reviewer1:

We have revised qPCR figure (now Figure 8 in revised version) using O. sativa actin as the reference gene. The relative quantities of each candidate gene transcript in the samples were calculated via 2-ΔCq [ΔCq = (Cq for candidate gene) – (Cq for actin)] method (Schmittgen and Livak, 2008; Nature Protocols 3:1101-1108). Obtained value in mock inoculated at 0 days were transformed into 1 and with identical conversion factor expression value of mock and nematode-inoculated at 7 days was transformed and statistically compared. Similar qPCR analysis method was used by Warmerdam et al. 2018, New Phytologist 218:724-737; Warmerdam et al. 2019, Molecular Plant Pathology 20:137-152.

As warranted by the esteemed reviewer we have now added a supplementary table depicting the expression stability of reference gene (actin) in different experimental conditions such as leaf, root, stem and seed tissues of resistant and susceptible genotype in mock and nematode inoculated condition. qRT-PCR is changed to RT-qPCR in the text of revised MS.

Line 219. Gel of RNA is given in supplementary figure S.

Line 220. Starting amount of RNA was 500 ng.

Response to Reviewer 3:

1) We have described the Relationship between resistant accessions and genotypes of significant SNPs in supplementary figure 5.

2) p.8, l. 226. Suggested change is effected in the revised MS.

3) p.11, l. 311-312. Suggested change is effected in the revised MS. 

4) p.12, l. 337-338. Indeed, population I, II and III should be designated as “Pro-Aus”, “Pro-Indica” and “Mid-Gangetic”, respectively. We have rectified the mistake in revised MS.

5) p.14, l. 394. Suggested change is effected in the revised MS.

6) p.17, l. 431. Suggested change is effected in the revised MS. 

7) p.18, l. 443. Suggested change is effected in the revised MS.

8) p.20, l. 523-525. The sentence conveys that since RRKN is not a pest of japonica rice, it has not undergone any selection process to inherit nematode resistance. Thus, japonica rice might have shown nematode susceptibility in our study.

9) p.22, l. 557. Suggested change is effected in the revised MS.

---

## [Editor Report · Decision Letter 1]

31 Aug 2020

A genome-wide association study in Indian wild rice accessions for resistance to the root-knot nematode Meloidogyne graminicola

PONE-D-20-22147R1

Dear Dr. Uma Rao,

We’re pleased to inform you that your manuscript has been judged scientifically suitable for publication and will be formally accepted for publication once it meets all outstanding technical requirements.

Kind regards,

Ji-Zhong Wan

Academic Editor

PLOS ONE
---

## [Editor Report · Acceptance letter]

9 Sep 2020

PONE-D-20-22147R1 

A genome-wide association study in Indian wild rice accessions for resistance to the root-knot nematode *Meloidogyne graminicola*

Dear Dr. Rao:

I'm pleased to inform you that your manuscript has been deemed suitable for publication in PLOS ONE. Congratulations! Your manuscript is now with our production department. 

Kind regards, 

on behalf of

Dr. Ji-Zhong Wan 

Academic Editor

PLOS ONE